# Mapping the path to domestic surrogacy: Identifying key facilitators and barriers in the Netherlands

Malene Tanderup[1,2*], Lone Schmidt[3], Amrita Pande[4], Birgitte Bruun Nielsen[5], Charlotte Kroløkke[6], Peter Humaidan[1,2]

1 Department of Clinical Medicine, Aarhus University, Denmark, 2 The Fertility Clinic Skive, Skive Regional Hospital, Denmark, 3 Department of Public Health, University of Copenhagen, Denmark, 4 Department of Sociology, University of Cape Town, South Africa, 5 Department of Gynaecology, Fertility and Childbirth, Copenhagen University Hospital, Copenhagen, Denmark, 6 Department of Culture and Language, University of Southern Denmark, Odense, Denmark

* mts@clin.au.dk

## Abstract

### Background

Surrogacy involves a woman who consents, before conception, to carry and deliver a child for individuals or couples unable to do so due to biological or medical limitations. This complex process encompasses medical, ethical, legal and financial considerations, resulting in varied legislation worldwide, with countries either prohibiting, restricting or legalising it. Recently, several nations have revised their legislation to encourage domestic surrogacy over international options, driven by ethical considerations and legal concerns. However, these revisions are still pending enactment. Despite the extensive literature addressing the legal, ethical, societal and medical challenges and benefits of surrogacy, no study has comprehensively analysed these factors together to fully capture the complexity of surrogacy implementation. This study aims to identify the key elements that currently facilitate the implementation of domestic surrogacy in the Netherlands and those essential elements needed for its successful continuation.

### Methods

A qualitative case study was conducted, employing both interviews and document analysis. The selection targeted individuals who were directly involved in or had an informed perspective on handling surrogacy in the Netherlands, including healthcare professionals, healthcare system leaders, policymakers, non-governmental organisations (NGOs), academics, lawyers and counsellors and 14 experts were purposively selected. The data were analysed both inductively and deductively, using the Context

**Data availability statement:** The data underlying this study cannot be shared publicly due to the protection of the privacy of individuals participating in the study. Anonymised data will be shared upon reasonable request to the corresponding author or the Danish Data Protection Agency (2016-051-000001, running number 2468) at fortegnelse@au.dk.

**Funding:** PH received a grant from the Independent Research Fund Denmark (grant number: 1030-00041B), which covers MT's employment as a PhD fellow and travel costs to the Netherlands. The funders had no role in study design, data collection and analysis, decision to publish, or preparation of the manuscript.

**Competing interests:** The authors have declared that no competing interests exist.

and Implementation of Complex Interventions (CICI) framework to assess the contextual factors influencing the implementation of domestic surrogacy.

## Results

Four CICI domains were identified as most influential on the implementation of surrogacy: legal (allowance of altruistic gestational surrogacy but missing legal framework on legal parentage, advertisement and payment), political (political shifts and experts' influence, gatekeepers, intersectional collaborations), ethical (professionals' influence on patient's choice) and socio-cultural (donation culture and public opinion). The absence of a legal framework that secures legal parenthood, the limited availability of fertility services and the shortage of surrogate candidates represent key barriers to the implementation of domestic surrogacy in the Netherlands. Conversely, significant facilitators include extensive, well-organised collaboration between professionals and non-governmental organisations (NGOs), invited by the political system to share expert knowledge and support comprehensive legislation.

## Conclusion

In conclusion, despite the progress achieved, domestic surrogacy remains largely inaccessible to most infertile individuals and is yet to be fully adopted. Without legal reforms, the situation of surrogacy in the Netherlands is likely to remain unchanged, mirroring the experiences of other countries with pending surrogacy legislation.

## Introduction

Surrogacy involves a woman, referred to as a surrogate, who agrees before pregnancy to carry and deliver a child for an individual or couple, referred to as the intended parent(s). This method of family building is becoming increasingly prevalent, allowing individuals or couples to attain parenthood under circumstances where carrying a child is biologically impossible or medically contraindicated [1]. However, despite its advantages, surrogacy involves a range of legal, medical, cultural, and ethical considerations, including legal parenthood, kinship, reproductive rights and justice, social responsibility, reproductive politics, and economics [2–6]. The legal landscape of surrogacy is complex and dynamic, with countries either restricting or opening their markets. The existence of disparate legislative frameworks has led some couples to seek surrogacy services abroad, referred to as transnational surrogacy [5,7,8].

The use of transnational surrogacy, and particularly the fluctuation of markets, has heightened the risks faced by intended parents, surrogates, and children born through these arrangements [9–13]. From the perspective of surrogates, especially in lower-income countries, issues such as isolation, stigmatisation and exploitation have been explored [14–18]. Nonetheless, this vulnerability is not one-dimensional; it can also be interpreted through the empowerment framework enabling women to, e.g., support their families [4,6].

In contrast, research on surrogates in Canada, the United States (US), and the United Kingdom (UK) suggests that these women experience greater societal acceptance and openness regarding their roles [19,20]. This positive environment often fosters closer relationships between surrogates and intended parents, particularly when they share the same national origin [19]. Contributing factors include the absence of language barriers, shared cultural understandings, and shorter physical distances [9,19,21,22]. Contributing factors include the absence of language barriers, shared cultural understandings, and shorter physical distances. As a result, several countries are revising their legislation to promote domestic surrogacy and reduce the need for international services [23,24]. However, there is a scholarly gap in the literature concerning the successful implementation of surrogacy, particularly in understanding the contextual factors involved. Therefore, our study aims to analyse these issues collectively to develop a comprehensive understanding of the complexities involved in implementing surrogacy. To explore the contextual factors affecting implementation, we chose to use the Context and Implementation of Complex Interventions (CICI), which is based on concept analysis and provides broad coverage of contextual issues [25]. The CICI framework describes seven contextual domains: geographical, epidemiological, socio-cultural, socio-economic, ethical, political and legal [26]. Furthermore, it offers a framework for evaluating the setting, context and implementation of complex interventions embedded, all within a broad public health perspective [26].

Understanding the terminology of surrogacy is crucial for practical implementation analysis as the categories provide insights into the logistical, ethical and legal complexities involved in the implementation process. Surrogacy can be categorised by fertility method, relationship, economics and geography. Medically, surrogacy is divided into traditional and gestational methods. Traditional surrogacy involves home insemination or assisted insemination using the intended father's sperm, making the surrogate both the genetic and the birthing mother. Gestational surrogacy employs *in vitro* fertilisation (IVF) at a fertility clinic, allowing the use of the intended mother's oocytes (eggs) and the intended father's sperm to create an embryo, which is then implanted in the surrogate. If the intended mother cannot provide oocytes, or in cases involving same-sex male couples or single men, an oocyte donor is also needed. Surrogacy can be known or unknown. In known surrogacy, the surrogate is a family member or close friend. Unknown surrogacy involves a surrogate who is initially a stranger to the intended parents, typically facilitated through an agency or online groups.

Economically, surrogacy is considered altruistic when no payment beyond expenses and minor compensation is made. Altruistic gestational surrogacy is legal in the Netherlands, the UK, Canada and Australia. Conversely, a few countries, including some states in the US, Ukraine, Georgia and Colombia, permit compensation beyond mere expenses [22,27–30]. In general, the service of gestational surrogacy is limited to the country's national or legal residents, while few countries including Canada, states in the US, Colombia, Mexico, Ukraine and Georgia extends availability to all [31].

The group utilising surrogacy is small but diverse, including women without a uterus due to congenital absence or removal for medical reasons, women with non-functional uteri, women for whom pregnancy is life-threatening, same-sex male couples and single men [32].

Interesting, representing a partially publicly reimbursed altruistic gestational surrogacy model, the Dutch setting has received little attention. In the Netherlands, fertility services in general are highly used [33], yet domestic gestational surrogacy remains rare. Between 1997 and 2017, only 95 couples underwent IVF surrogacy, resulting in 50 births, despite over 500 applications [34,35]. However, it is estimated that 150 Dutch couples seek surrogacy services abroad on an annual basis [36]. Conversely, in the UK, domestic altruistic surrogacy is relatively common. Of the more than 500 children born annually through surrogacy in the UK, half are born via domestic arrangements [9,37]. Infertile individuals from the UK seeking surrogacy abroad primarily opt for the US, India and Ukraine [9]. Between 2014 and 2020, 21,649 children were born through gestational surrogacy in the US [38], where economic compensation to the surrogate is permissible. In 2013, 18.5% of these births involved non-US residents [39]. Surrogate cycles thus constituted 4% of all fertility cycles in the US, comparable to Canada's rate of 2.3% in 2022 [40], yet markedly contrasting with the Netherlands' rate, which is below 0.05% (own calculation) [33,35].

Countries such as Canada, Ireland, Belgium, New Zealand and the UK are comparable to the Netherlands in their use of altruistic gestational surrogacy. While Canada, New Zealand and the UK have established legal frameworks regulating surrogacy practices, Ireland, Belgium and the Netherlands have laws that do not explicitly prohibit altruistic gestational surrogacy [5,19,41]. Notably, all these countries except Canada are currently revising their legislation to provide a comprehensive legal framework for surrogacy to make domestic surrogacy more accessible and encourage intended parents to opt for domestic solutions, thereby reducing the need for transnational surrogacy [5,24,41]. In contrast, Canada is experiencing an increasing rate of domestic surrogacy, which may be attributed to a robust legal framework that permits the use of agencies, allows advertising to find a surrogate, and promotes non-discriminatory practices for accessing IVF services [19]. Furthermore, countries such as Denmark, Iceland and Finland, which do not currently offer gestational surrogacy, have recently reevaluated whether to introduce it. However, they have been reluctant to do so due to ethical, legal and organisational considerations [42–45].

No publications have examined facilitators and barriers to implementing domestic altruistic gestational surrogacy. This study aims to identify key elements currently in place to support the implementation process of domestic surrogacy in the Netherlands, along with those necessary for its successful continuation, through an in-depth analysis of interviews and documents, using the CICI framework. Two research questions guided this study.

1) How do key stakeholders within the clinic, including doctors and psychologists, as well as those in the organisation, such as counsellors, lawyers, and officials, experience the surrogacy process?

2) What are the barriers preventing the expansion of domestic surrogacy, and which factors have shaped its development to its current state?

## The Netherlands as a case study

In 1985, gynaecologist Sylvia Dermout was diagnosed with gynaecological cancer, leading her to become the chairperson of the Dutch gynaecological cancer organisation Olijf [46]. In this role, she championed the cause of post-cancer women seeking to have genetically linked children through surrogacy. She sought to exert influence through political, medical and psychological channels. In 1994, the Dutch Minister of Health, Dr. Els Borst, amended the law to allow non-commercial IVF surrogacy under strict conditions set by the Dutch Society of Obstetrics and Gynaecology (NVOG). This led to the establishment of the Dutch Centre for Non-commercial IVF Surrogacy in Zaandam, led by Dr. Dermout, where a study from 1997 to 2004 resulted in the birth of 16 children through altruistic gestational surrogacy [34]. The Netherlands was one of the first countries worldwide to introduce altruistic gestational surrogacy in public facilities; until then, gestational surrogacy was known only from private fertility services, since 1985 in the US and the UK [47,48].

A consultation board, comprising six gynaecologists, a professor of psychology, a professor of ethics and a lawyer, reviewed all surrogacy cases to ensure that clinical indications were met. After the pilot study, existing fertility centres were hesitant to assume clinical responsibility for surrogacy. In 2006, with state funding, the Free University Hospital (VUmc) continued the programme, though without Dr. Dermout. It took years to reach similar case numbers as in Zaandam [35]. In 2018, the NVOG revised its guidelines, changing the requirement from the child needing to be fully genetically related to the intended parents to requiring at least one genetically related intended parent. This amendment facilitated the use of gestational surrogacy by same-sex couples [49]. In 2019, the specialised fertility clinic Nij Geertgen began offering surrogacy, becoming the second facility in the Netherlands to do so. Known for treating many same-sex female couples, they aimed to make surrogacy available for same-sex male couples as well [50]. A pending law proposal seeks to establish a legal framework for surrogacy, aiming to shift from discouraging to promoting domestic surrogacy and reducing the number of individuals pursuing surrogacy abroad. The initiative began in 2012 when the Parliament, in response to numerous legal changes in family law, requested an expert panel to prepare a report on family law focused on the child's best interests. This culminated in the so-called 2016 report, which forms the foundation of the current law proposal. The organisational and political events are illustrated in Fig 1.

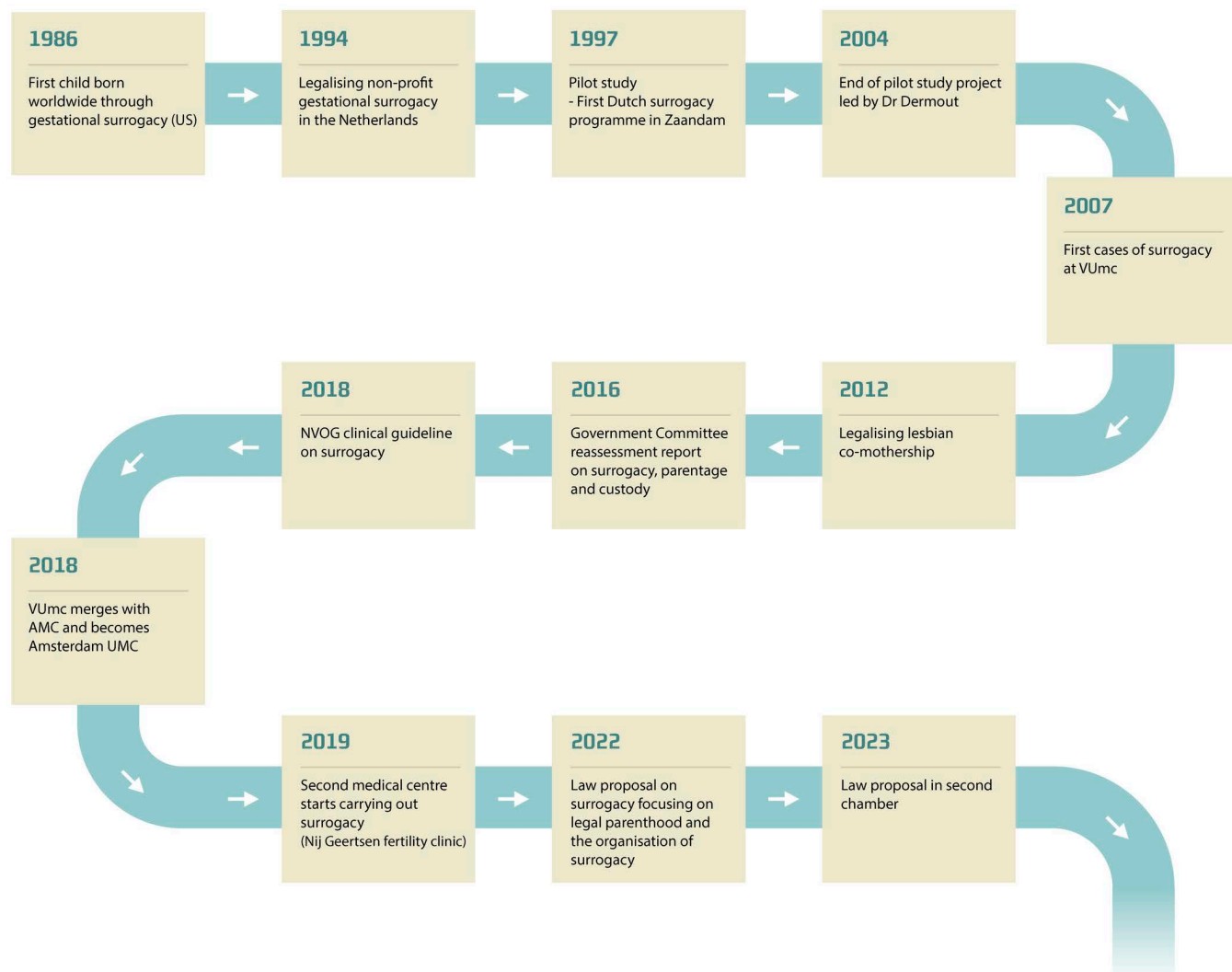

**Fig 1. Chronology of the intervention.** Key historical events leading to the introduction of gestational surrogacy in the Netherlands. VUmc: the Free University Hospital; NVOG: the Dutch Society of Obstetrics and Gynaecology; AMC: Amsterdam Medical Centre.

The Dutch healthcare system rests on a universal social health insurance model combining public oversight with private insurance. All residents are required to obtain statutory health insurance, which is provided by private insurers obliged to offer coverage to all. The system is financed primarily through public funds derived from premiums, taxes and government contributions. Regarding reproductive health, the system provides varying degrees of coverage for family planning and infertility treatments. Basic health insurance covers certain services, including IVF, in the event of medical necessity. Reimbursement for gestational surrogacy is partial, covering solely the cost of embryo creation as the transfer of the embryo to a fertile woman is not covered. The system provides coverage for medically indicated cases only and excludes reimbursement of IVF expenses for same-sex male couples. All fertility services are conducted in public fertility departments and clinics [51, chapter 2 pp 65–67].

## Methods

### Study design

The research adopted a qualitative methodology, integrating in-depth interviews with 14 key informants conducted between March and May 2024, alongside document analysis. This approach was designed to investigate the evolution and implementation of domestic gestational surrogacy in the Netherlands, tracing its history from its inception in 1990 to the pivotal year of 2024. This period marks a significant legislative milestone as reforms of existing surrogacy laws are currently under review by the second chamber of the Dutch Parliament. This methodology facilitated an in-depth examination of the perspectives and experiences of a diverse range of professionals. These individuals are either directly engaged in leading roles in the coordination of surrogacy arrangements or play a crucial role in the shaping of the legislative and regulatory framework that governs surrogacy practices.

### Key informant interviews

The material includes interviews with 14 purposively selected experts who were central to understanding the organisation and development of surrogacy in the Netherlands. The selection targeted individuals who were directly involved in or had an informed perspective on handling surrogacy, including healthcare professionals, healthcare system leaders, policymakers, non-governmental organisations (NGOs), academics, lawyers and counsellors (Table 1). Half of the interviewed stakeholders were, besides their professional career with surrogacy, also in an NGO with an interest in surrogacy. Twelve of the 14 interviewed had influenced the formation of previous and current law proposals, either as advisers in the Ministry of Justice and the Ministry of Health or in writing the 2016 report. The stakeholders had nine years of experience with surrogacy on average. Given the small number of people involved in surrogacy, and to ensure that confidentiality is maintained, the information provided on the profile of key informants is purposefully limited. Where possible and available, the research team asked key informants for documentary evidence to support claims made during the interview.

### Data collection

A semi-structured interview guide was developed with open-ended questions examining surrogacy practices in the Netherlands, focusing on the institutional and cultural contexts in which surrogacy takes place. The interviews were conducted in

**Table 1. Key informant interviews by profession and experience with surrogacy.**

| Personnel type | Years of experience with surrogacy |
|---|---|
| Healthcare professional | 14 |
| Healthcare professional | 14 |
| Counsellor | 6 |
| Counsellor | 11 |
| Healthcare professional | 9 |
| NGO | 14 |
| Healthcare professional | 5 |
| Lawyer | 17 |
| Healthcare professional | 10 |
| Lawyer | 15 |
| Academia | 5 |
| Academia | 12 |
| Healthcare official/policy maker | 2 |
| Academia | 2 |

English by the first author (a medical doctor), as it is widely understood in the Netherlands, allowing for a fluid and natural conversation. Conducting the interviews in English instead of Dutch eliminated the need for a translator, which could have hindered the flow of discussion and potentially introduced misunderstandings or nuances lost in translation. However, this choice also represents a compromise, as it may limit the depth of expression for some interviewees. The interviews were audio-recorded, pseudonymised and transcribed by the first author. The duration of the interviews was, on average, 1h25min (50min to 2h33min). NVivo14 was employed for data management and analysis.

## Document analysis

A document analysis was conducted. We focused solely on Dutch policy documents and official statements pertaining to surrogacy. This included the initial description of a national policy on surrogacy from 1990, as well as the latest updated national law proposal on domestic and transnational surrogacy from 2024. It is important to note that we did not include analyses or comments on the law itself, nor on any associated legal interpretations. Additionally, relevant academic literature was examined to contextualise the findings within the broader discourse surrounding surrogacy. For an overview of included documents, see Table 2.

## Data analysis

The analytical procedure unfolded over two phases, adopting a combined inductive-deductive methodology. During the initial phase, we employed a tailored version of Malterud's systematic text condensation as an empirical method for coding the interviews and documents [53]. This phase was instrumental in generating codes that laid the groundwork for the subsequent phase. The initial phase encompassed: (i) reading the entire material to gain an overall impression and identify preliminary themes, (ii) identifying codes related to the preliminary themes, (iii) condensation of the codes to meaning and (iv) forming descriptions and concepts. The second phase was steered by the CICI framework [26] with a particular focus on the contextual factors influencing the use of domestic surrogacy. Codes were systematically categorised under

**Table 2. Analysed documents influencing the field of surrogacy in the Netherlands.**

| No | Name and content | Type | Responsible for the document | Year of publication |
|----|------------------|------|------------------------------|---------------------|
| 1 | 21968, 3<br>Discourage surrogacy but not banning it.<br>Ban on mediators/brokers to prevent commercial surrogacy | Amendment of the Criminal Code with some provisions aimed at countering commercial surrogacy | Minister of Justice, E.M.H. Hirsch Ballin | 1990-1991 |
| 2 | 25000, XVI, 51<br>Idealistic surrogacy at IVF Centre | Recommendation to second chamber | Minister of Health, E. Borst | 1997 |
| 3 | 25000, XVI, 54<br>Legal parentship: the surrogate mother is the legal mother.<br>By court, legal adoption can be given to the intended parents. | Recommendation to second chamber | Minister of Health, E. Borst | 1997 |
| 4 | "*De eerste logeerpartij: hoogtechnologisch draagmoederschap in Nederland*"<br>The first sleepover: gestational surrogacy in the Netherlands [46] | PhD thesis | Sylvia Dermout's PhD thesis | 2001 |
| 5 | Child and Parents in the 21st Century<br>(called "the 2016 report") [52] | Report on reassessment on parenthood | Government Committee | 2016 |
| 6 | Chapter 4 "Surrogacy" Modelreglement Embryowet [49] | Clinical guidelines on surrogacy | NVOG – the Netherlands Society of Obstetrics and Gynaecology | 2018 |
| 7 | Draft legislation surrogacy, the Netherlands | Law proposal | Ministry of Justice | 2024 |

the analytic domains: geographical, epidemiological, socio-cultural, socio-economic, ethical, legal and political (see Fig 2). This framework provided a methodological and targeted analytical lens.

The methodological triangulation of key informant interviews and document analysis, coupled with the diligent application of the CICI framework, produced comprehensive insights into the facilitators and barriers of using surrogacy within the Dutch setting amidst possible legislative changes. The following section presents the findings of this study.

### Ethical approval

Under the Dutch Medical Research with Human Subjects Law (WMO), research involving interviews that do not require participants to undergo procedures or adhere to specific behavioural rules is exempt from approval. This exemption also applies to the institutional ethical review board at Aarhus University, Denmark. Approval for this study was granted by the Danish Data Protection Agency (2016-051-000001, running number 2468). The study complied with the Helsinki II Declaration, and informed oral consent was obtained from all interviewees prior to conducting the interviews. Additionally, written confirmation of participation was received via individual emails from the experts.

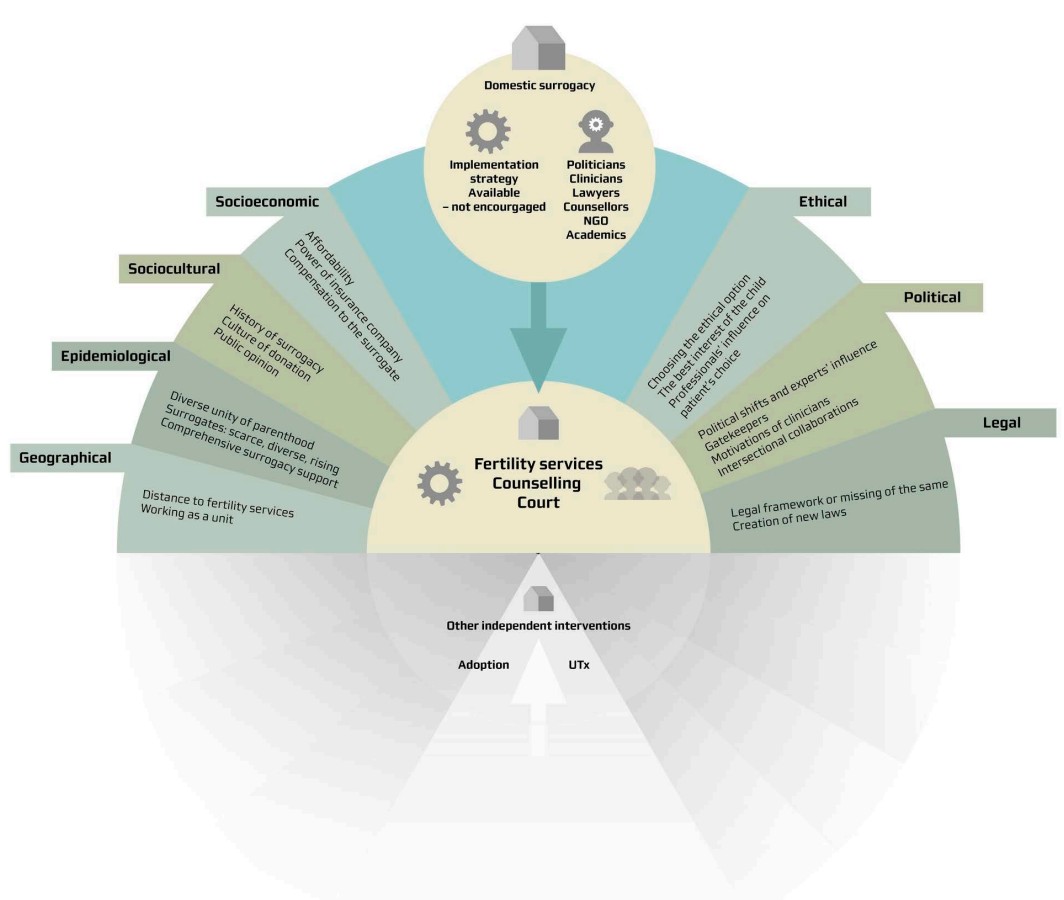

**Fig 2. CICI Framework: Illustration of the context, implementation and setting involved in domestic surrogacy in the Netherlands.** UTx: Uterus transplantation.

## Inclusivity in global research

Additional information regarding the ethical, cultural, and scientific considerations specific to inclusivity in global research is included in the Supporting Information (S1 File)

## Preconceptions

MT, PH, LS, and BBN are medical doctors working in fertility units and obstetric wards. We recognise that our personal perceptions of conception, pregnancy, and childbirth are evolving, especially as we compare societal norms with our professional experiences. Our medical backgrounds have equipped us with a profound understanding of the trauma experienced by both women and men on their journey to parenthood.

Together with CK and AP, we have been involved in extensive research related to surrogacy for over a decade. This includes conducting fieldwork in India among surrogates and professionals, as well as studying Danish intended parents who utilise transnational commercial surrogacy. These experiences have provided us with a nuanced perspective on surrogacy, enhancing our understanding of its complexities. We have observed the diverse motivations and challenges faced by individuals, encompassing the emotional and ethical dilemmas surrounding surrogacy, as well as the socio-cultural implications that impact families.

## Results

The interview data predominantly aligned with four out of the seven domains of the CICI framework: 1) legal, 2) political, 3) sociocultural and 4) ethical. Within these categories, distinct themes emerged during the interviews. These themes, referred to hereafter as the context's sub-domains [26], are characterised by a distinction between elements already present and those identified by the interviewees as necessary. Moreover, the interviewees were able to identify the key elements currently in place to support the implementation process of domestic surrogacy, along with those essential for the successful continuation of this process. For a comprehensive overview of the domains, sub-domains and key elements, please refer to Table 3.

Table 3. Existing and required elements within each subdomain of legal, political, ethical and socio-cultural domains.

| Domain | Sub-domain | Key elements | |
|---|---|---|---|
| | | Existing | Required |
| Legal | | Permission of altruistic gestational surrogacy | Legal framework on legal parentage, advertisement and payment |
| Political | Political shifts and experts' influence | Proactive involvement of experts in the political system | Continued progress of law proposal |
| | Gatekeepers | Healthcare professionals set criteria for IVF surrogacy | |
| | | Court and some cases Council for Child Protection as gatekeepers after birth | Council for Child Protection and Court as pre-conceptional gatekeeper |
| | Motivations of clinicians | High value of offering surrogacy | More clinics offering surrogacy through legal framework and higher tariffs |
| | Intersectional collaboration | Teamwork and short lines | |
| Ethical | Choosing the ethical option | Professionals' influence on patient's choice | |
| Socio-cultural | Culture of donation | Professionals carry the Dutch culture | Bank of surrogate candidate |
| | | Strong value placed on child's origin story | |
| | Public opinion | Media coverage of surrogacy | |

 

## Legal domain

The legal context plays a pivotal role in surrogacy. The first law in 1990 aimed to discourage commercial surrogacy, both domestically and transnationally (document 1). The removal of the ban on fertility clinics assisting in surrogacy in 1997 (document 2) paved the way for the first pilot test of altruistic surrogacy in the Netherlands [34]. Since then, no changes have been made to the law. The lack of a comprehensive legal framework, with no specific requirements or rules for the surrogacy process, is seen by the interviewees as a major limitation to the use of domestic surrogacy.

Three requirements are considered crucial for tailoring the legal framework to improve domestic surrogacy: 1) legal parentage from birth, 2) advertisement and 3) financial compensation. Document 5 highlights the current legal uncertainty:

*"This legal uncertainty ensures that, at the moment of birth, it is unclear who the legal parents of the child will be, which surname the child will have and, in some cases, which nationality the child possesses. It also means that the persons who will be caring for and raising the child, in the majority of cases, are not (or at least not yet) the persons who are able to make important decisions regarding the child. Until the end of the surrogacy arrangement, the surrogate and the intended parents are extremely interdependent on each other, as well as on the Child Protection Board and ultimately the court."*

The law proposal addresses these issues. Currently, Dutch law prohibits advertising to find a surrogate mother or intended parents and acting as a mediator in finding a surrogate mother. Although current technology, such as closed Facebook groups, makes it possible for people to bypass these restrictions, the ban still has an effect. As one NGO stated: *"There are still people who hold back because it is still illegal. When that ban is lifted, people will more easily find each other".*

A counsellor echoed this when stating: *"They like to stay in the Netherlands because it costs a lot of money to go abroad. But it's sometimes too difficult here. You may not advertise, so how do you find the surrogate mother? That's very difficult".*

The legal framework prohibiting advertisement also includes a ban on agencies or brokers. However, many interviewees view professional assistance in the matching process as essential. Professionals help ensure thorough screening of all parties and match personalities based on expectations regarding topics such as abortion, communication and involvement in the child's life.

In gestational surrogacy arrangements, healthcare professionals oversee the screening process for the surrogate and the intended parents. However, in traditional surrogacy, the interviewees were clear that there is no obligation to seek professional counselling. This means that intended parents risk selecting a surrogate with potential "red flags," such as psychiatric issues or a complicated personality. An NGO representative noted, *"When you have your pink glasses on and are dreaming of a baby, it is hard to be strict about the choice of the potential surrogate".* Conversely, some interviewees pointed out that surrogates may struggle to reject couples, fearing they might shatter their dreams if the chemistry does not feel right. The law proposal suggests the introduction of a non-profit, non-government agency to facilitate this process. However, no organisation has yet shown interest in taking on this role.

The interviews made it clear that there is no explicit cap on financial compensation. However, the general prohibition on child trafficking and child buying means that commercial surrogacy is not practised. It is permitted to reimburse the surrogate mother and the oocyte donor for expenses incurred, and in addition, they can receive 190€ per month, which is the amount a volunteer in the Netherlands can obtain. Interviewees clearly expressed that the law should not introduce commercial surrogacy in the Netherlands as it is not "part of Dutch culture" and is not in the "best interest of the child" to risk being commodified. In addition to this the Government Committee report suggested a symbolic compensation of 500€ per month (doc 5). However, the latest version of the law proposal, which only allows for reimbursement of expenses without any extra compensation, was viewed as a limitation of domestic surrogacy. One NGO representative suggested that a small payment would help increase the number of domestic surrogates:

[6]:*"I don't mean big amounts as in America, but the Government Committee advice was 500€ a month…I was really charmed by the idea that no one would do it to get rich quickly – there are easier ways – but it means - it is substantial…To get the kids to a theme park because the kids have been neglected a little bit because you have been sleeping more. Things like that".*

In this manner, the legal framework plays a dominant role in shaping domestic surrogacy arrangements, creating barriers, at times, regarding legal uncertainty of parenthood, the recruitment of surrogates and their potential compensation.

## Political domain

The political domain seeks to regulate and optimise the distribution of power, resources and interests among citizens. It also addresses the various organisations involved, their objectives and the official and unofficial regulations that govern their interactions. Additionally, this domain encompasses the healthcare system and the accessibility of its services [26].

Political shifts and experts' influence.  All interviewees mentioned that the advancement of the law proposal is largely contingent on the political priority assigned to family law. In the autumn of 2023, the political landscape shifted to the right, favouring national conservative parties. Since the interviews were conducted, a right-wing government was finally formed after seven months. Most interviewees were sceptical that the law proposal would pass the second of three rounds in the House of Representatives (second chamber).

In 2012, the Parliament requested an expert panel to draft a report on family law, focusing on the child's best interests. The appointed Government Committee, tasked with evaluating the desirability of laws on surrogacy, multi-parent custody, and intentional multi-parenting, consisted of experts in family law, medical ethics, fertility law, gynaecology and obstetrics, children's law, private and international law, pedagogy and sociology. The report, entitled "Child and Parents in the 21st Century" and commonly referred to as "the 2016 report", had a significant impact on the current law proposal on surrogacy (document 5, [52]). Interviewees frequently mentioned this report. One policy-maker noted:

*"From then on, every time the government made a regulation in this area, and you asked, why are you making this regulation - oh it's because it's in the report. Because it came from a commission with a lot of experts appointed by the government. It had a lot of influence…2016 was like a shift point."*

As one of the experts involved in drafting the report said, when the report was published, public reaction surprisingly focused on multi-parent custody rather than surrogacy. Ultimately, the Government declined the proposal for multi-parent custody, allowing political work to continue with surrogacy.

In developing gestational surrogacy legislation, the political system has proactively involved various stakeholders, including clinicians, counsellors, academics, lawyers and NGOs. Interviewees expressed satisfaction with their participation from the outset and during the drafting of the 2016 report and law proposals. The Ministry of Justice continues to consult these stakeholders for advice on surrogacy issues. This close relationship is regarded as essential for ensuring that the law prioritises the child's best interests and is workable for both domestic and international surrogacy cases. However, the future development of the legal framework is dependent on the political climate and remains susceptible to changes due to shifts in political regimes and public opinions.

Gatekeepers.  In the current system, the type of surrogacy chosen determines which gatekeepers couples and surrogates will encounter. For domestic gestational surrogacy, the primary gatekeeper is the medical doctor, who assesses the medical indication for surrogacy. However, in the Netherlands, fertility law stipulates only age limitations, whereas the NVOG set out clinical guidelines that are obliged to be followed. However, indications for using surrogacy are loosely specified criteria, such as a non-functioning uterus with intact ovaries. The clinicians, regarded as experts, have the discretion to establish their own criteria for offering treatment. Consequently, the two clinics that provide surrogacy

services in the Netherlands may interpret conditions like a non-functioning uterus differently, and for a couple of years, only one of the clinics offered its service to same-sex male couples.

Many intended parents halt the process early, either due to indications deemed insufficient for surrogacy or the lack of a potential surrogate. Clinicians note that the majority discontinue on their own after receiving information about the process, while others are formally rejected. An NGO representing surrogates and intended parents noted:

> "It is very difficult to get into the clinics. There are only two clinics, and they are only picking the ones with the best story and the best chances...If you potentially can get pregnant yourself, then you are not eligible, and there are thousands of reasons why they wouldn't be admitted."

The exclusion criteria for surrogate candidates varied between clinics. One clinic considered obstetric risks, such as a previous Caesarean section, as grounds for exclusion. The other argued that, after thorough examination by an experienced obstetrician and receiving extensive information, the surrogate should make her own decision.

Psychologists conduct screenings to assess the motivations and expectations of surrogates and intended parents. This step varies between clinics, with some psychologists being internal and others external. Few candidates are rejected at this stage, mainly due to unclear motivations.

Before starting IVF treatment, the involved parties are also advised to consult a lawyer experienced in surrogacy, of whom there are few in the Netherlands. Although these lawyers are external to the clinics, they liaise with them to ensure the legal feasibility of proceeding with the surrogacy case. The legal process also commences during pregnancy to guarantee that Children's Board authorises the intended parents to take the infant home following birth and to shorten the proceeding for the intended parents to become the child's legal parents. A court ruling issued one year post-birth, can conclude the legal parentage status of the intended parents.

Despite an increase in the number of surrogacy cases over the past five years, the two clinics handle only 25–30 annually. In contrast, one of the interviewed lawyers handles over 40 cases annually, with 73% involving surrogacy abroad. Professionals in the legal and counselling fields note that approximately half of the couples using a domestic option choose traditional surrogacy, particularly same-sex male couples who do not require an oocyte donor if the surrogate uses her own gametes. This method reduces waiting time and bypasses gatekeepers. Moreover, it can decrease the number of individuals involved in the child's creation, thereby limiting the number of people they may need to engage with later. This simplification results in a less complex origin story for the child. However, many traditional surrogacy cases remain uncounted as they do not involve professionals.

Seeking legal or counselling support before conception or during pregnancy facilitates the subsequent parentage or adoption process. Nevertheless, traditional surrogacy lacks the medical, psychological and legal screening of intended parents and surrogates. Interviewees reported that the few cases with negative outcomes were those where no preconception professional assistance was sought.

To ensure a smooth process for both traditional and gestational surrogacy, one lawyer emphasised the importance of following a carefully planned process:

> "In general, I would say it is super important that people follow the careful process to succeed. Take time to get to know each other, do the counselling, get the proper legal work done, insurance—everything. Then we also see that the procedures, gestational or traditional, national or international, go quite well."

**A doctor explained why some couples avoid gestational surrogacy**

> "They don't want to go to the hospital because of the costs and slower procedures. You have long waiting times. You don't know which doctor you will get, all these questions, all these things. And as a gay couple, you have your surrogate and can start immediately at home."

One suggestion in the law proposal is to establish the primary legal gatekeeper before conception for all cases, whether traditional, gestational, national, or international. This would require all parties to undergo counselling and be checked by the RvdK before conception.

Motivations of the clinicians.  When discussing facilitators and barriers to surrogacy implementation, it is crucial to consider clinicians' motivations. Interviewees frequently cite the limited number of clinics as a significant barrier. Only two fertility services in the Netherlands offer surrogacy, and both leaders have stated that economic factors, often presumed to be the primary motivation, are not actually the driving forces. One clinician stated, *"It is not something you do to earn money. You cannot earn money from it… it is not easy money. It is a special population. Especially the surrogate mothers are extraordinary women. It makes my job interesting. That's it."*

Another clinician added, *"You don't carry out surrogacy to make money… What we gain is marketing and a good feeling… everyone with a desire for a child should be able to fulfil it, regardless of sexual orientation or diseases."*

The clinics are allowed to charge a maximum tariff of €2500 for an IVF surrogacy treatment. This tariff is the same for an ordinary IVF, although a surrogacy IVF requires more scans, coordination and consultations. The vulnerability of smaller clinics in offering surrogacy was also highlighted as there is a risk that, in the event of a problematic case, the head doctor could be sued by authorities, potentially leading to the closure of the entire clinic. Few are willing to take this risk, especially without any economic incentive and transparent legal framework.

Intersectional collaboration.  Intersectional collaboration is one of the key elements facilitating domestic surrogacy in the Netherlands. Interviewees repeatedly emphasised its paramount importance in managing complex processes like surrogacy. This collaboration occurs at various levels and is centred on clear communication, proximity and mutual knowledge within the field.

Doctors work closely with psychologists to discuss complex cases. In clinical settings, psychologists can refer patients to a counsellor or psychologist closer to the patient's location, avoiding long travel distances, such as from the eastern part of the Netherlands to Amsterdam. Doctors and psychologists also refer to lawyers with in-depth knowledge of surrogacy and vice versa. Lawyers update healthcare professionals on recent legal developments or clarify legal situations in specific cases. The interdisciplinary collaboration between counsellors and lawyers is also emphasised.

One interviewee noted, *"They [intended parents or surrogates] intend to go to a lawyer when there are difficulties, and when the lawyers think they need counselling, they send them to us. So it's important that there are short lines between the lawyers and the counsellors and that we know each other."*

NGOs working with intended parents and surrogates advise people on who to contact in different situations, and many professionals are also part of these NGOs. The collaboration between NGOs and professionals has been enhanced by the government-supported development of an information webpage on surrogacy. An NGO representative mentioned, *"It started with the surrogacy information homepage ordered by the government. They gathered people from clinics, counsellors and NGOs, including us, which allowed me to meet them in person."*

Furthermore, personal involvement and collaboration are evident within the professional community. For instance, lawyers meet at least twice yearly to discuss ethical boundaries. One lawyer explained the necessity of the meetings, *"Where do we draw the line ethically? Do we want to do this? What do we think? It's important for developments, and we keep each other updated".*

The political domain benefits from key facilitators, including a small but proactive community of professionals actively engaged with the political system. The collaboration among gatekeepers and other professionals has played an essential role in ensuring the safety of surrogacy procedures in the Netherlands, both traditional and gestational. Another existing element is the motivation of clinicians and other professionals to engage in surrogacy, which appears to be driven not primarily by financial incentives, but rather ideological and professional considerations. However, the interviewees highlighted the need for more clinics offering surrogacy, which could be encouraged by establishing a clear framework and setting specific tariffs for surrogacy services. Another requirement for advancing domestic surrogacy is the introduction

of a proposed primary legal gatekeeper. This role would ensure that all parties receive comprehensive information on the legal, ethical and medical aspects of surrogacy, and an understanding of the advantages and disadvantages of using domestic surrogacy over transnational surrogacy.

### Ethical domain

<u>Choosing the ethical option</u>.  It has been surprising to discover how strongly the ethical perspective on third-party reproduction, particularly surrogacy, is held among NGOs, lawyers, counsellors and clinicians and how much it influences the choices of intended parents. All interviewees favoured domestic over international surrogacy, arguing that it was in the best interest of both the child and the surrogate. They mentioned factors such as the origin story, connectedness, communication and ensuring informed consent as reasons for prioritising a domestic solution. These factors are also closely related to factors from the sociocultural domain.

If domestic surrogacy is not chosen, intended parents are strongly advised to opt for surrogacy in the US or Canada rather than in Eastern European or South American countries. One way to enforce this ethical framework could be to forbid travelling abroad or restrict certain countries. However, a policymaker noted: *"In an ideal world, you might forbid it and people wouldn't do it, but that is not how it works. So, they go abroad."*

Instead, the existing approach relies on the moral stance of the people involved in the organisation, who limit their services to intended parents based on certain choices. For example, one of the lawyers said: *"We do see in some countries that surrogate mothers, egg donors and the interests of the child are not well taken care of. Personally, I have decided to no longer assist in those cases."* Consequently, the lawyer's practice was limited to cases conducted in the Netherlands, the US or Canada, a pattern observed in most lawyers' practices.

An NGO representative echoed this sentiment: *"If someone says, 'I want to go to Georgia,' then I say, 'That might not be a good idea because of this and this and this.' Maybe that person doesn't change their mind, but hopefully, others who are not as far along in their process will be influenced."*

In addition to ethical considerations, the legal context significantly impacts couples considering surrogacy in countries like Ukraine and Georgia. For instance, a child born through surrogacy in Ukraine is recognised by the Ukrainian law as the child of the Dutch parents. However, the Dutch court does not acknowledge this way of becoming a legal parent, despite the genetic link to the Dutch parents. Consequently, the child must appear in a Dutch court to obtain Dutch citizenship. This process is complicated because the child cannot travel to the Netherlands without a Dutch passport, potentially delaying the child's entry into the Netherlands by over a year.

To prevent the occurrence of stateless children and ensure that ethical considerations and Dutch perspectives on surrogacy are addressed, a new law proposal recommends mandatory counselling in the Netherlands. The purpose of the law proposal, explained by a policymaker, is: *"To really push them to do it carefully, but we are putting the responsibility on them [intended parents]. We will not completely forbid it but stimulate them to do it here."*

A Dutch national ethical board does not exist; instead, the former appointed Government Committee, which drafted the 2016 report (doc 5), formulated the ethical perspectives surrounding surrogacy in the Netherlands. These perspectives are reflected in the practices and views on surrogacy held and carried out by professionals in the field and thus affect the users of surrogacy.

### Socio-cultural domain

The socio-cultural domain comprises explicit and implicit behaviour patterns and shared ideas and values within a group. It encompasses the conditions of life, social roles and relationships. This context includes knowledge, beliefs, customs and institutions [26]. The historical context of gestational surrogacy in the Netherlands is outlined in the introduction, providing a foundation for understanding the current surrogacy landscape in the Netherlands. To maintain a clear focus on

the key issues affecting the implementation of surrogacy, we have focused on the 'culture of donation' and 'public opinion' even though the socio-cultural domain in surrogacy is extensive.

Culture of donation.  The dearth of women willing to serve as surrogates in the Netherlands has been identified by all the interviewees as a significant obstacle to the utilisation of domestic surrogacy. A legal framework securing the surrogate's interest is one way to encourage women to consider entering surrogacy. Another important perspective is a country's culture of third-party reproduction, including gamete donation and surrogacy. In the Netherlands, anonymous gamete donation has been prohibited since 2004, based on the assumption that it is in the child's best interest to know its origin story, as also formulated in the 2016 report. One of the lawyers who handles most of these cases has indicated that although approximately half of her clients opt for gestational surrogacy abroad, they are fully aware of the legal and ethical necessity for non-anonymous donation:

*"In my practice, it only happens now and then that people have used an anonymous donor, and mostly those are expats. Most Dutch intended parents are very well aware of the fact that using a non-anonymous donor is required by legislation that has been in place for 20 years. There have been many television programmes about children searching for their donors, and there were some scandals in recent years about mass donors, which have been widely covered in the news."*

The media play a role in informing the public about charitable giving norms in the Netherlands. The legal context also complicates the adoption process for a child born through surrogacy when anonymous donation is used.

The majority of individuals initially seek to utilise family members as gamete donors to maintain their genetic lineage. However, there is a shortage of all types of donors in the Netherlands. Some parents elect to utilise sperm from Danish companies despite these companies' less than exemplary reputation in the Netherlands. The potential difficulties associated with utilising a donor or surrogate abroad were elucidated in Document 5, drawing upon the jurisprudence of the European Court of Human Rights: *If treatment is not possible or provided for in the Netherlands, it is possible to use such services abroad. In these situations, there are often fewer, and in some cases no, possibilities for these children to ascertain their origin story.* Counsellors and NGOs emphasise the advantages of gamete donation and surrogacy in the Netherlands for the child's future. An NGO representative said: *"What I always tell the intended parents is that the bigger the distance you have to the surrogate or egg donor, the bigger distance your child has to cross to meet their roots. You make it more difficult for your kid when they want to know where they come from, what is my story."*

Using a known surrogate simplifies the child's origin story, increasing the likelihood that the child will have a closer relationship with the surrogate, who might be an aunt or a family friend. Another way to simplify the origin story is through gestational surrogacy using the intended mother's oocyte, or traditional surrogacy, where the oocyte donor and surrogate are the same person. This approach is prevalent among same-sex male couples in the Netherlands.

To encourage more women to become surrogates, it is suggested that robust legal frameworks and the establishment of a surrogate bank, like a sperm bank, could be essential. A surrogate bank would register and screen potential surrogates, ensuring that they meet the necessary criteria.

Public opinion.  The interviewees noted that, in general, the public does not pay much attention to the legal intricacies of surrogacy. Media coverage tends to focus more on the consequences of anonymous donations. One researcher in the field explained that, according to her survey on public opinion regarding surrogacy, the majority of the 1,000 participants had a positive attitude towards it, particularly when it involved a heterosexual couple with a known surrogate.

The interviewees also expressed concern that scandalous cases involving surrogacy or an excessive focus on LGBT rights, particularly the right to use surrogacy, could have a negative impact on public opinion. Such a shift could potentially halt the practice of surrogacy and impede the progress of related legal proposals.

In the socio-cultural domain, the culture surrounding donation has affected the shortage of both oocyte donors and surrogates and has been a major barrier to domestic surrogacy. The high value placed on the possibility for the child to know his or her origins is a facilitator for more professionals to guide couples to choose a domestic option instead of a foreign one. Necessary elements include a legal framework to protect the interests of surrogates, and a culture supportive of third-party reproduction could encourage more women to participate. Creating a surrogate bank to register and screen potential surrogates is also suggested as a solution.

## Discussion

Through an in-depth analysis of interviews and documents using the CICI framework, we have identified facilitators and barriers affecting the implementation of domestic gestational surrogacy in the Netherlands. This analysis explores the legal, political, ethical and socio-cultural contexts to elucidate the existence of specific barriers and facilitators. Key barriers include the absence of a legal framework securing legal parenthood and political reluctance to enact the proposed legislation. Additionally, there are limited fertility clinics willing to offer surrogacy services, as it is complex task with no economic incentives. Furthermore, the shortage of surrogate candidates is exacerbated by restrictions that prohibit advertisements and the involvement of agencies. Conversely, significant facilitators encompass extensive, well-organised collaboration between professionals and NGOs invited by the political system to share expert knowledge and support comprehensive legislation.

Our findings underscore the complexity of implementing surrogacy in a global north setting, even over a 27-year period due to the interplay of various contextual determinants. While these determinants have been described individually in multiple research studies [5,9,54], to our knowledge, no study has specifically aimed to analyse them collectively to understand the full complexity of implementing surrogacy.

The legal domain represents a crucial prerequisite for further implementation, as evidenced by all interviewees' responses: the introduction of a comprehensive legal framework is essential. This is in line with the notable shift in legislative objectives, which initially discouraged surrogacy but made it accessible to a limited population and is now focused on promoting domestic surrogacy to reduce the number of individuals seeking surrogacy abroad. The framework should address the legal aspects of parenthood, guarantee the intended parents' and the surrogate's rights, facilitate the advertising process to enable parties to find each other and provide financial compensation for the surrogate. The necessity of such a legal framework is not exclusive to the Netherlands; comparable developments are occurring in the UK, Ireland and Australia, of which none of these have yet been enacted [5,41,54].

Legal factors are closely tied to political dynamics. The recent shift from a left-wing to a right-wing government has cast uncertainty on the enactment of the proposed law. The vulnerability to political dynamics is also seen in Spain [55] and Finland, where a political election halted the introduction of new surrogacy legislation [56]. The driving force behind the law proposal has been the incorporation of medical, legal, psychological and patient-oriented perspectives over several years. Interviewees felt that being heard at the ministerial level was significant during the development of the Committee Report (2016) and the subsequent law proposal. This involvement may have prevented a scenario similar to that in Iceland, where the law proposal faced overwhelming criticism from various interest groups, including medical and social worker associations and human rights organisations, ultimately leading to no changes in the legal situation in Iceland [42]. However, as noted within the legal domain, new legislation is essential for the broader adoption of domestic surrogacy in place of transnational surrogacy.

The CICI analysis also highlighted the role of gatekeepers in the organisation of surrogacy. Current gatekeepers set criteria for gestational surrogacy, with healthcare professionals screening intended parents and surrogates. However, declined intended parents or surrogates or those never seeking consultations can still pursue transnational gestational or domestic traditional surrogacy. Legal parentage and stepchild adoption are decided by the court and the Council of Child Protection, but these institutions' decisions occur after the child is born, limiting their impact on surrogacy practices.

A national gatekeeper in the form of preconception counselling of all types of surrogacy is suggested as a required element to enhance domestic surrogacy. Instituting a gatekeeper function may increase the number of intended parents choosing domestic surrogacy by highlighting its advantages over international options. If implemented, avoiding long waiting times and reducing travel distances are crucial. Besides providing valuable information to intended parents and surrogates, it is proposed that intended parents and surrogates approved during the consultation process achieve legal parenthood from the child's birth. This would represent a significant improvement compared to the current prolonged uncertainty surrounding the establishment of legal parenthood. Conversely, individuals who opt out of preconception consultations would still face the existing lengthy adoption procedures, involving years of legal procedures. A similar regulatory body is included in the new Irish surrogacy law proposal, which will preapprove surrogacy arrangements and introduce medical and capacity checks [5]. The effectiveness of centralised counselling services in improving and encouraging domestic surrogacy remains to be seen.

Furthermore, existing Dutch gatekeepers have requested a clear legal framework to make more clinics comfortable and safe in offering surrogacy as a fertility treatment. The NVOG recommends limiting surrogacy to one to two clinics nationally due to its complexity [49]. However, interviewees believe that only two clinics are insufficient to meet infertile couples' needs and vulnerable to changes in case of critical personal circumstances of professionals or shifting management in the clinics.

Additionally, the study shows the importance of intersectional collaboration for the feasibility and safety of intended parents and surrogates. Unlike commercial surrogacy in countries like Ukraine, the US and Colombia, the Netherlands does not offer a "full surrogacy package" with integrated services from lawyers, clinicians, agents, counsellors and psychologists. Instead, each profession operates independently, potentially creating abrupt and insecure processes. However, the cohesiveness of the processes has been strengthened by informal collaborations built over the years, particularly since the 2016 report, and the launch of a government-supported information homepage. This collaboration influences the ethical domain, particularly the sub-theme of "choosing the ethical option". Interviewees expressed strong opinions on ethical options and aimed to guide the choices of intended parents and surrogates in the direction of domestic solutions.

A recent survey reveals generally positive perceptions of surrogacy within the Dutch population, particularly for heterosexual couples utilising a Dutch surrogate who is a friend or family member [57]. Supportive attitudes, previous studies show, are more likely in countries where surrogacy is legal, and they affect both the surrogate and the intended parents [10,14,19,44,58,59]. Despite the positive attitude in the Netherlands, the shortage of oocyte donors and surrogates remains a significant limiting factor. Key elements identified to address this issue include the ability to advertise for potential surrogates, the establishment of a "surrogate bank" akin to sperm and oocyte banks, and the implementation of a comprehensive legal framework for surrogacy. The Canadian experience with altruistic gestational surrogacy indicates that permitting advertising for surrogates could significantly increase the pool of candidates; hence, a Canadian survey showed that 93% of surrogates met the intended parents through the Internet or an agency [19]. Moreover, the issue of economic compensation for surrogates was highlighted by interviewees and also mentioned in the 2016 report. While this compensation is not akin to that found in commercial surrogacy, it is more of a symbolic payment comparable to altruistic oocyte and sperm donation, recognising the surrogate's contribution. The suggested compensation would be €500 per month; however, there remains uncertainty about which expenses can be covered, as this is not clearly defined in the current Dutch framework. The Canadian model allows for reimbursement of all expenses, including clothing, food, medical costs, childcare, travel expenses, and lost wages. This ensures that surrogates do not experience any financial losses during their pregnancies, a situation that contrasts sharply with the current practices in the Netherlands.

Contrary to our initial assumption that countries permitting gestational surrogacy would not utilise traditional models, our findings suggest otherwise. Our research reveals a notable, albeit unquantifiable, use of traditional surrogacy in the Netherlands, previously unknown outside the country. In nations where gestational surrogacy is prevalent, traditional surrogacy typically remains rare [60,61] as the American Society of Reproductive Medicine (ASRM) advocates for gestational

surrogacy [62], and some countries even prohibit traditional surrogacy [63,64]. The preference for medicalisation to uphold professionalism and prevent strong bonds between the surrogate and the genetically linked child leads intended parents to use different oocyte donors from the surrogate if they cannot provide the oocytes themselves [65]. However, within the Dutch context, professionals reported no significant differences between traditional and gestational surrogacy in terms of conflicts or complications, provided thorough counselling and alignment of expectations were in place before conception. This finding aligns with studies conducted in the US and the UK [66,67]. Traditional surrogacy, with its lower cost, reduced medical risks and fewer individuals involved in the child's origin story, warrants further research to explore its potential as an alternative to gestational surrogacy. The relatively high utilisation of traditional surrogacy in the Dutch setting, coupled with the increase in cases observed in clinics over recent years, indicates that the implementation of surrogacy is evolving. Even so, it is not yet fully established. Hence, interviewees estimate that more than half of the intended parents still seek surrogacy services abroad.

## Strengths and limitations

This study offers a comprehensive examination of the implementation process of domestic surrogacy through the lens of the CICI framework. It involved a diverse range of professionals engaged in surrogacy practices within the Netherlands, including clinicians, private counsellors, and lawyers. This comprehensive approach allows for an investigation into domestic traditional surrogacy experiences, which have been inadequately addressed in Dutch publications due to the absence of such cases in the clinical setting..

However, some methodological considerations must be taken into account when interpreting the findings. The interviews were limited to healthcare and legal professionals actively involved in surrogacy, many of whom have been part of the development of previous or current surrogacy law proposals, which introduces the potential for confirmation bias. However, the proposed legislation has not yet been enacted due to the current political climate, leading experts to critically discuss the implications of the existing system and the limitations that may persist even if the proposed law is eventually passed. To further mitigate the risk of confirmation bias, including professionals who have opted not to handle surrogacy cases would have been beneficial. However, meaningful insights from such individuals would require them to deliberately choose not to engage in surrogacy work, as those who have not participated may lack comprehensive knowledge of the field. It is important to note that the factors identified are based on the interviewees' perceptions, which may be shaped by their contexts. However, these insights provide a deep understanding of the dynamics present in the Netherlands. Many of these factors have also been identified in studies from other countries, suggesting that they could potentially be generalised to other national contexts.

Additionally, a significant limitation is the absence of surrogates and intended parents as interviewees. Including their perspectives could have highlighted facilitators and barriers to the organisation of surrogacy from the user viewpoint. However, a Dutch research group is concurrently conducting interviews with Dutch intended parents and surrogates, making it ethically redundant to replicate this study. Insights from their research are expected to contribute valuable knowledge to this field.

## Conclusion

The CICI framework has been instrumental in identifying existing elements and areas requiring further development to enhance the implementation of surrogacy in the Netherlands, aiming to favour domestic surrogacy over transnational alternatives. Despite progress, domestic surrogacy remains largely inaccessible and is still in the process of being fully adopted.

A crucial step towards further implementation is establishing a comprehensive legal framework, which hinges on governmental engagement and prioritisation. Such a framework would provide greater legal security for intended parents and encourage more women to become surrogates by ensuring their rights and safety. Without these legal changes, surrogacy in the Netherlands will likely remain unchanged, like in other countries with pending legal reforms.

 

Our findings offer valuable insights for healthcare system policymakers, decision-makers and professionals involved in domestic surrogacy, both within the Netherlands and in countries undergoing similar processes.

## Supporting information

**S1 File. Inclusivity in global research.** Checklist.
(DOCX)

## Acknowledgments

We would like to thank all the healthcare professionals, lawyers, policy-makers, academics and counsellors who took the time to participate in the interviews. Furthermore, we would like to thank Mia Fredens and Stine Bollerup at the Institute of Public Health, Defactum, at Aarhus University, for providing valuable feedback on earlier versions of this manuscript.

## Author contributions

**Conceptualization:** Malene Tanderup, Lone Schmidt, Amrita Pande, Birgitte Bruun Nielsen, Charlotte Kroløkke, Peter Humaidan.

**Data curation:** Malene Tanderup.

**Formal analysis:** Malene Tanderup.

**Funding acquisition:** Peter Humaidan.

**Investigation:** Malene Tanderup.

**Methodology:** Malene Tanderup, Lone Schmidt, Amrita Pande, Charlotte Kroløkke.

**Project administration:** Malene Tanderup.

**Software:** Malene Tanderup.

**Supervision:** Lone Schmidt, Amrita Pande, Birgitte Bruun Nielsen, Charlotte Kroløkke, Peter Humaidan.

**Validation:** Malene Tanderup.

**Visualization:** Malene Tanderup.

**Writing – original draft:** Malene Tanderup.

**Writing – review & editing:** Malene Tanderup, Lone Schmidt, Amrita Pande, Birgitte Bruun Nielsen, Charlotte Kroløkke, Peter Humaidan.

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
