## [Decision Letter · Decision Letter 0]

PONE-D-24-39971Navigating the complexities of domestic surrogacy: A qualitative case study from the NetherlandsPLOS ONE

Dear Dr. Tanderup,

Thank you for submitting your manuscript to PLOS ONE. After careful consideration, we feel that it has merit but does not fully meet PLOS ONE’s publication criteria as it currently stands. Therefore, we invite you to submit a revised version of the manuscript that addresses the points raised during the review process.

Key Areas for Improvement:

Work on the concerns raised by the reviewers in the introduction section (reviewers #1 and #2).Redraw Figure 1 and Figure 2 and make them much clearer (Reviewer #2).Kindly address the concerns raised by the reviewers in the discussion section of your work (reviewers #1 and #2).Address all the concerns raised by Reviewer #1 regarding the methods as well.

In summary, I encourage the authors to address all the reviewers' comments and make the necessary revisions, particularly in improving the introduction, methods, and discussion sections. I look forward to reviewing your revised manuscript.

Please submit your revised manuscript by Apr 04 2025 11:59PM. If you will need more time than this to complete your revisions, please reply to this message or contact the journal office at plosone@plos.org . Please include the following items when submitting your revised manuscript:

We look forward to receiving your revised manuscript.

Kind regards,

Godwin Banafo Akrong, Ph.D.

Academic Editor

PLOS ONE

2.  Please amend either the title on the online submission form (via Edit Submission) or the title in the manuscript so that they are identical.

3. Please include a complete copy of PLOS’ questionnaire on inclusivity in global research in your revised manuscript. Our policy for research in this area aims to improve transparency in the reporting of research performed outside of researchers’ own country or community. The policy applies to researchers who have travelled to a different country to conduct research, research with Indigenous populations or their lands, and research on cultural artefacts. The questionnaire can also be requested at the journal’s discretion for any other submissions, even if these conditions are not met.  Please find more information on the policy and a link to download a blank copy of the questionnaire here: https://journals.plos.org/plosone/s/best-practices-in-research-reporting. Please upload a completed version of your questionnaire as Supporting Information when you resubmit your manuscript.

4. In the ethics statement in the Methods, you have specified that verbal consent was obtained. Please provide additional details regarding how this consent was documented and witnessed, and state whether this was approved by the IRB.

 [PH received a grant from the Independent Research Fund Denmark (grant number: 1030-00041B), which covers MT’s employment as a PhD fellow and travel costs to the Netherland]. 

6.  We note that you have indicated that there are restrictions to data sharing for this study. For studies involving human research participant data or other sensitive data, we encourage authors to share de-identified or anonymized data. However, when data cannot be publicly shared for ethical reasons, we allow authors to make their data sets available upon request. For information on unacceptable data access restrictions, please see http://journals.plos.org/plosone/s/data-availability#loc-unacceptable-data-access-restrictions.

7. In the online submission form, you indicated that [The data underlying this study cannot be shared publicly due to the protection of the privacy of individuals participating in the study. Anonymised data will be shared upon reasonable request to the corresponding author.].

Reviewers' comments:

Reviewer's Responses to Questions

**Comments to the Author**

1. Is the manuscript technically sound, and do the data support the conclusions?

Reviewer #1: Yes

Reviewer #2: Yes

2. Has the statistical analysis been performed appropriately and rigorously? 

Reviewer #1: N/A

Reviewer #2: Yes

3. Have the authors made all data underlying the findings in their manuscript fully available?

Reviewer #1: No

Reviewer #2: No

4. Is the manuscript presented in an intelligible fashion and written in standard English?

Reviewer #1: Yes

Reviewer #2: Yes

5. Review Comments to the Author

Reviewer #1: Introduction: -

1. The introduction highlights ethical, legal, and socio-cultural issues which is useful. However, it does not explicitly introduce the CICI framework that is later used in the study. A brief mention of the theoretical basis for analyzing surrogacy policies would strengthen the study’s conceptual foundation.

2. The introduction focuses primarily on policymakers and intended parents but does not discuss surrogates’ perspectives. A more inclusive stakeholder discussion (e.g., surrogates, fertility clinics, NGOs, children born through surrogacy) would improve comprehensiveness.

3. The introduction provides the aim of the study but does not explicitly state any research questions. The authors should include clearly formulated research questions to help strengthen the study.

Methods: -

The qualitative study approach was adopted in the study with in-depth interviews and document analysis being used to explore the evolution and implementation of domestic surrogacy in the Netherlands. The use of triangulation provides a multi-faceted approach and makes the data for the study credible. Also the 14 key informants selected from diverse backgrounds allow the unearthing of perspectives from those involved in policy-making and legal structuring. The data analysis approach was also systematic

However, the authors need to take note of these gaps and address them

1. With regards to the participants selected for the study, the absence of surrogates and intended parents is a critical gap. Their voices would provide first-hand experiential insights rather than relying on professionals’ interpretations. The exclusion of their voices makes the study reflect more of institutional and policy-level perspectives than the lived experiences of those directly undergoing surrogacy. Including their voices would provide grounded, experiential perspectives, balancing the top-down policymaker insights.

2. 12 out of the 14 participants were involved in forming previous and current surrogacy law proposals, meaning most informants share a policymaker or expert perspective. This risks confirmation bias, as informants may reinforce existing legislative perspectives rather than challenging them.

3. The interviews were conducted in English, which is a second language for both the interviewer and interviewees. This could affect the depth of responses, particularly in nuanced legal and ethical discussions. Would it not have been possible to conduct interviews in Dutch and translate responses, reducing the risk of lost meaning in translation? Can the authors highlight the strengths of using English over Dutch in their interviews and justify their choice based on the concern raised?

4. The document analysis includes key policy documents but lacks a clear explanation of inclusion criteria. It is not clear whether opposing perspectives (e.g., critiques of Dutch surrogacy laws) were actively sought out which could have been the case. Can the authors clarify why certain policy documents were selected and whether contrasting viewpoints (e.g., legal critiques) were considered?

5. The study does not discuss the issue of reflexivity. Given the sensitive nature of surrogacy, acknowledging positionality and preconceptions would strengthen methodological transparency. The backgrounds and potential biases of the researchers should therefore be discussed. Providing a section on this will show the authors acknowledgement of potential biases in data interpretation.

Results and discussion

1. The findings are heavily reliant on institutional and policy-level informants as opposed to other participants in the surrogacy industry. (Refer to the methods section for further clarity)

2. While qualitative data is well analyzed, there is no numerical representation of themes (e.g., frequency of specific concerns among participants). The authors should consider including some descriptive statistics (e.g., how many participants mentioned a given challenge) to strengthen validity

3. The authors briefly reference the UK and other European models. However, a detailed comparison with countries where surrogacy policies are fully developed (e.g., Canada, the U.S.) would enhance policy recommendations

4. The study confirms well-documented barriers to surrogacy (e.g., legal uncertainties, lack of fertility services, and political hesitancy), but does not uncover radically new challenges or facilitators. Insights on this could be unearthed if point 3 is addressed.

Reviewer #2: I thank the authors for an interesting and original study. The method used is quite unique and I think it’s a good addition to the surrogacy literature.

My main critique pertains to the introduction – I think it would benefit from further review to ensure it provides the reader with all the necessary knowledge to understand the purpose and context of the study. For example:

• I would bring the definitions earlier so that you define the words before you use them. For example, right at the beginning of the introduction you explain the Netherlands permits altruistic gestational surrogacy but don’t explain what these words mean until later. This could be confusing to a reader not across the terminology.

• The risks of international surrogacy require more unpacking as it currently only explains “varying reproductive standards, legal uncertainty, the potential exploitation of women and the risk of commodifying children”. The risks of international surrogacy are the main driving force for promoting domestic surrogacy, and hence the rational for your study.

• The introduction would also benefit from an explanation of the recent law proposal. For example, what prompted it, what does it propose? What stage is it currently at? While it is mentioned in Figure 1, it doesn’t have any accompanying text and so its relevance is unclear.

• Consider moving the explanation of the public healthcare system (currently in the methods) to the introduction as it feels oddly placed in the methods.

Other more minor comments:

• Consider an alternative title which makes explicit reference to the purpose of the paper. The current title implies the paper is presenting an investigating into the lived experience of navigating domestic surrogacy.

• Typo in the abstract sentence “Despite there is extensive literature addressing the legal, ethical, societal and medical challenges and benefits of surrogacy”.

• Typo in the abstract sentence “…domestic surrogacy remains inaccessible largely inaccessible to…”

• Line 81 “British infertiles” seems a problematic description.

• Figure 1 is quite blurry and figure 2 is completely unreadable

• Line 398– what is the “origin story” and why is simplifying it a good thing?

• The discussion could benefit from a paragraph on surrogate compensation – the “hot topic” in surrogacy.

6. PLOS authors have the option to publish the peer review history of their article (what does this mean? ). If published, this will include your full peer review and any attached files.

**Do you want your identity to be public for this peer review?** For information about this choice, including consent withdrawal, please see our Privacy Policy .

Reviewer #1: No

Reviewer #2: No

---

## [Author Response · Author response to Decision Letter 1]

6 Apr 2025

Major revision of PONE-D-24-39971 entitled “Navigating the complexities of domestic surrogacy: A qualitative case study from the Netherlands”

Dear Editor in Chief and Associated Editor,

Thank you for allowing us to respond to the comments given by the reviewers. Below please find a structured response to all comments. We thank the reviewer for all the comments and questions, which have helped us improve the paper. We hope that you will find the answers and the rewritten text in line with your expectations.

1. Reviewer #1:

1.1. Introduction:

The introduction highlights ethical, legal, and socio-cultural issues which is useful.

1.1.1. However, it does not explicitly introduce the CICI framework that is later used in the study. A brief mention of the theoretical basis for analyzing surrogacy policies would strengthen the study’s conceptual foundation.

We agree and have moved the introduction of the CICI framework so it is in the beginning of the introduction section

1.1.2. The introduction focuses primarily on policymakers and intended parents but does not discuss surrogates’ perspectives. A more inclusive stakeholder discussion (e.g., surrogates, fertility clinics, NGOs, children born through surrogacy) would improve comprehensiveness.

We agree and have extended the introduction especially including the surrogates perspectives as well. However, it is not including all stakeholders, as the introduction section is already lengthy and they will be introduced in the results section and discussion more extensively. We do hope you will accept this prioritization.

Lines 75-89: The use of transnational surrogacy, and particularly the fluctuation of markets, has heightened the risks faced by intended parents, surrogates, and children born through these arrangements [9, 11-14]. From the perspective of surrogates, especially in lower-income countries, issues such as isolation, stigmatisation and exploitation have been explored [15-19]. Nonetheless, this vulnerability is not one-dimensional; it can also be interpreted through the empowerment framework enabling women to e.g. support their families [4, 6].

In contrast, research on surrogates in Canada, the United States (US), and the United Kingdom (UK) suggests that these women experience greater societal acceptance and openness regarding their roles [20, 21]. This positive environment often fosters closer relationships between surrogates and intended parents, particularly when they share the same national origin [20]. Contributing factors include the absence of language barriers, shared cultural understandings, and shorter physical distances [9, 20, 22, 23]. Contributing factors include the absence of language barriers, shared cultural understandings, and shorter physical distances [24, 25].

1.1.3. The introduction provides the aim of the study but does not explicitly state any research questions. The authors should include clearly formulated research questions to help strengthen the study.

We agree that the article would benefit with the addition of the research questions behind the study.

Lines 155-160: Two research questions guided this study.

1) How do key stakeholders within the clinic, including doctors and psychologists, as well as those in the organisation, such as counsellors, lawyers, and officials, experience the surrogacy process?

2) What are the barriers preventing the expansion of domestic surrogacy, and which factors have shaped its development to its current state?

1.2. Methods:

The qualitative study approach was adopted in the study with in-depth interviews and document analysis being used to explore the evolution and implementation of domestic surrogacy in the Netherlands. The use of triangulation provides a multi-faceted approach and makes the data for the study credible. Also the 14 key informants selected from diverse backgrounds allow the unearthing of perspectives from those involved in policy-making and legal structuring. The data analysis approach was also systematic.

However, the authors need to take note of these gaps and address them

1.2.1. With regards to the participants selected for the study, the absence of surrogates and intended parents is a critical gap. Their voices would provide first-hand experiential insights rather than relying on professionals’ interpretations. The exclusion of their voices makes the study reflect more of institutional and policy-level perspectives than the lived experiences of those directly undergoing surrogacy. Including their voices would provide grounded, experiential perspectives, balancing the top-down policymaker insights.

We agree that including surrogates and intended parents would have significantly enriched the study. Initially, we planned to interview these groups to identify facilitators and barriers in the organisation of surrogacy from a user perspective. However, during the PhD period of the first author, another study focusing on the experiences of intended parents and surrogates in the Netherlands was initiated by two Dutch PhD students. Given this overlap, it was deemed ethically redundant to conduct a similar investigation.

Once their results are published, they will provide valuable insights that can be discussed alongside the findings of our study, which primarily presents an institutional and policy-level perspective. We have acknowledged this limitation in the manuscript's revised section.

Added to Strengths and limitation section

Lines 777-782: Additionally, a significant limitation is the absence of surrogates and intended parents as interviewees. Including their perspectives could have highlighted facilitators and barriers to the organisation of surrogacy from the user viewpoint. However, a Dutch research group is concurrently conducting interviews with Dutch intended parents and surrogates, making it ethically redundant to replicate this study. Insights from their research are expected to contribute valuable knowledge to this field.

1.2.2. 12 out of the 14 participants were involved in forming previous and current surrogacy law proposals, meaning most informants share a policymaker or expert perspective. This risks confirmation bias, as informants may reinforce existing legislative perspectives rather than challenging them.

We agree that this matter needs to be addressed in the article, and it has now been incorporated into the strengths and limitations section.

Lines 763-772: The interviews were limited to healthcare and legal professionals actively involved in surrogacy, many of whom have been part of the development of previous or current surrogacy law proposals, which introduces the potential for confirmation bias. However, the proposed legislation has not yet been enacted due to the current political climate, leading experts to critically discuss the implications of the existing system and the limitations that may persist even if the proposed law is eventually passed. To further mitigate the risk of confirmation bias, including professionals who have opted not to handle surrogacy cases would have been beneficial. However, meaningful insights from such individuals would require them to deliberately choose not to engage in surrogacy work, as those who have not participated may lack comprehensive knowledge of the field.

1.2.3. The interviews were conducted in English, which is a second language for both the interviewer and interviewees. This could affect the depth of responses, particularly in nuanced legal and ethical discussions. Would it not have been possible to conduct interviews in Dutch and translate responses, reducing the risk of lost meaning in translation?

Can the authors highlight the strengths of using English over Dutch in their interviews and justify their choice based on the concern raised?¨

We agree that the choice of language used in the interviews should be described. The decision to conduct the interviews in English, which is widely understood in the Netherlands, was made to facilitate a more fluid and natural conversation. While we acknowledge that conducting interviews in Dutch could have provided deeper insights into nuanced legal and ethical discussions, our experience during the interviews did not indicate a limitation in this regard. Participants expressed themselves comfortably in English, which allowed for rich dialogue.

Lines 235-240: The interviews were conducted in English by the first author (a medical doctor), as it is widely understood in the Netherlands, allowing for a fluid and natural conversation. Conducting the interviews in English eliminated the need for a translator, which could have hindered the flow of discussion and potentially introduced misunderstandings or nuances lost in translation. However, this choice also represents a compromise, as it may limit the depth of expression for some interviewees

1.2.4. The document analysis includes key policy documents but lacks a clear explanation of inclusion criteria. It is not clear whether opposing perspectives (e.g., critiques of Dutch surrogacy laws) were actively sought out which could have been the case. Can the authors clarify why certain policy documents were selected and whether contrasting viewpoints (e.g., legal critiques) were considered?

We agree that clarification regarding our document analysis is essential. We will clarify our inclusion criteria of the documents.

Lines 245-251: We focused solely on Dutch policy documents and official statements pertaining to surrogacy. This included the initial description of a national policy on surrogacy from 1990, as well as the latest updated national law proposal on domestic and transnational surrogacy from 2024. It is important to note that we did not include analyses or comments on the law itself, nor on any associated legal interpretations. Additionally, relevant academic literature was examined to contextualise the findings within the broader discourse surrounding surrogacy.

1.2.5. The study does not discuss the issue of reflexivity. Given the sensitive nature of surrogacy, acknowledging positionality and preconceptions would strengthen methodological transparency. The backgrounds and potential biases of the researchers should therefore be discussed. Providing a section on this will show the authors acknowledgement of potential biases in data interpretation.

We agree that including preconceptions is important; however, this section was initially excluded to limit the length of the article. We have now added the section to address these insights.

Lines 273-285: MT, PH, LS, and BBN are medical doctors working in fertility units and obstetric wards. We recognise that our personal perceptions of conception, pregnancy, and childbirth are evolving, especially as we compare societal norms with our professional experiences. Our medical backgrounds have equipped us with a profound understanding of the trauma experienced by both women and men on their journey to parenthood.

Together with CK and AP, we have been involved in extensive research related to surrogacy for over a decade. This includes conducting fieldwork in India among surrogates and professionals, as well as studying Danish intended parents who utilise transnational commercial surrogacy. These experiences have provided us with a nuanced perspective on surrogacy, enhancing our understanding of its complexities. We have observed the diverse motivations and challenges faced by individuals, encompassing the emotional and ethical dilemmas surrounding surrogacy, as well as the socio-cultural implications that impact families.

1.3. Results and discussion

1.3.1. The findings are heavily reliant on institutional and policy-level informants as opposed to other participants in the surrogacy industry. (Refer to the methods section for further clarity)

We agree and this has been commented in 1.2.1.

1.3.2. While qualitative data is well analyzed, there is no numerical representation of themes (e.g., frequency of specific concerns among participants). The authors should consider including some descriptive statistics (e.g., how many participants mentioned a given challenge) to strengthen validity.

Thank you for your valuable feedback. The decision not to include numerical representations of themes in the analysis was deliberate, as numerical values are not typically employed in systematic text condensation or the CICI model. We have included statements indicating when the majority of interviewees comment on a particular theme; however, even if a theme is mentioned by only a few participants, it can still hold significant value. The richness of the qualitative data and its relevance to the context are far more meaningful than reducing the findings to mere numerical values. We believe this approach provides a more nuanced understanding of the participants' experiences.

1.3.3. The authors briefly reference the UK and other European models. However, a detailed comparison with countries where surrogacy policies are fully developed (e.g., Canada, the U.S.) would enhance policy recommendations

We agree that the Canadian case should be prominently highlighted as an example of altruistic gestational surrogacy, characterised by a high prevalence of domestic surrogacy and a robust legal framework. While we have included some examples of legislation in the US, we have refrained from making a detailed comparison due to the considerable variability in laws across different states

Lines 144-147: In contrast, Canada is experiencing an increasing rate of domestic surrogacy, which may be attributed to a robust legal framework that permits the use of agencies, allows advertising to find a surrogate, and promotes non-discriminatory practices for accessing IVF services.

Canada is also included in the discussion section, with additional emphasis placed on the economic compensation of surrogates, using Canadian practices as a key example.

Lines 722-734: The Canadian experience with altruistic gestational surrogacy suggests that allowing advertising for a surrogate could increase the number of surrogate candidates; hence, a Canadian survey showed that 93% of surrogates met the intended parents through the Internet or an agency [19].

The Canadian model allows for reimbursement of all expenses, including clothing, food, medical costs, childcare, travel expenses, and lost wages. This ensures that surrogates do not experience any financial losses during their pregnancies, a situation that contrasts sharply with the current practices in the Netherlands.

1.3.4. The study confirms well-documented barriers to surrogacy (e.g., legal uncertainties, lack of fertility services, and political hesitancy), but does not uncover radically new challenges or facilitators. Insights on this could be unearthed if point 3 is addressed.

We agree and have extended the first part of the discussion section

Lines 640-645: Key barriers include the absence of a legal framework securing legal parenthood and political reluctance to enact the proposed legislation. Additionally, there are limited fertility clinics willing to offer surrogacy services, as it is complex task with no economic incentives. Furthermore, the shortage of surrogate candidates is exacerbated by restrictions that prohibit advertisements and the involvement of agencies.

2. Reviewer #2:

I thank the authors for an interesting and original study. The method used is quite unique and I think it’s a good addition to the surrogacy literature.

2.1. Introduction

My main critique pertains to the introduction – I think it would benefit from further review to ensure it provides the reader with all the necessary knowledge to understand the purpose and context of the study. For example:

2.1.1. I would bring the definitions earlier so that you define the words before you use them. For example, right at the beginning of the introduction you explain the Netherlands permits altruistic gestational surrogacy but don’t explain what these words mean until later. This could be confusing to a reader not across the terminology.

We agree that the definitions should absolutely come before using the words. We have rearranged the introduction section to meet this comment.

2.1.2. The risks of international surrogacy require more unpacking as it currently only explains

---

## [Decision Letter · Decision Letter 1]

Mapping the path to domestic surrogacy: identifying key facilitators and barriers in the Netherlands

PONE-D-24-39971R1

Dear Dr. Tanderup,

We’re pleased to inform you that your manuscript has been judged scientifically suitable for publication and will be formally accepted for publication once it meets all outstanding technical requirements.

Kind regards,

Godwin Banafo Akrong, Ph.D.

Academic Editor

PLOS ONE

Additional Editor Comments (optional):

Reviewers' comments:

Reviewer's Responses to Questions

**Comments to the Author**

1. If the authors have adequately addressed your comments raised in a previous round of review and you feel that this manuscript is now acceptable for publication, you may indicate that here to bypass the “Comments to the Author” section, enter your conflict of interest statement in the “Confidential to Editor” section, and submit your "Accept" recommendation.

Reviewer #1: All comments have been addressed

Reviewer #3: All comments have been addressed

2. Is the manuscript technically sound, and do the data support the conclusions?

Reviewer #1: (No Response)

Reviewer #3: Yes

3. Has the statistical analysis been performed appropriately and rigorously? 

Reviewer #1: (No Response)

Reviewer #3: Yes

4. Have the authors made all data underlying the findings in their manuscript fully available?

Reviewer #1: (No Response)

Reviewer #3: Yes

5. Is the manuscript presented in an intelligible fashion and written in standard English?

Reviewer #1: (No Response)

Reviewer #3: Yes

6. Review Comments to the Author

Reviewer #1: (No Response)

Reviewer #3: The authors have thoroughly addressed all the comments and suggestions from the previous reviewers. I appreciate their revisions and the improvements made to the manuscript, which have significantly enhanced the clarity and quality of their work

7. PLOS authors have the option to publish the peer review history of their article (what does this mean? ). If published, this will include your full peer review and any attached files.

**Do you want your identity to be public for this peer review?** For information about this choice, including consent withdrawal, please see our Privacy Policy .

Reviewer #1: No

Reviewer #3: **Yes: ** Neema Landey

---

## [Editor Report · Acceptance letter]

PONE-D-24-39971R1

PLOS ONE

Dear Dr. Tanderup,

I'm pleased to inform you that your manuscript has been deemed suitable for publication in PLOS ONE. Congratulations! Your manuscript is now being handed over to our production team.

Kind regards,

on behalf of

Dr. Godwin Banafo Akrong

Academic Editor

PLOS ONE